# Weakly Supervised Ternary Stream Data Augmentation Fine-Grained Classification Network for Identifying Acute Lymphoblastic Leukemia

**DOI:** 10.3390/diagnostics12010016

**Published:** 2021-12-22

**Authors:** Yunfei Liu, Pu Chen, Junran Zhang, Nian Liu, Yan Liu

**Affiliations:** 1Department of Automation, College of Electrical Engineering, Sichuan University, Chengdu 610065, China; liuyunfei@stu.scu.edu.cn (Y.L.); junranzhang@scu.edu.cn (J.Z.); liu-nian@scu.edu.cn (N.L.); 2Department of Laboratory Medicine, Zhongshan Hospital Fudan University, Shanghai 200032, China; chen.pu@zs-hospital.sh.cn

**Keywords:** acute lymphoblastic leukemia, data augmentation, fine-grained classification, convolutional neural network

## Abstract

Due to the high incidence of acute lymphoblastic leukemia (ALL) worldwide as well as its rapid and fatal progression, timely microscopy screening of peripheral blood smears is essential for the rapid diagnosis of ALL. However, screening manually is time-consuming and tedious and may lead to missed or misdiagnosis due to subjective bias; on the other hand, artificially intelligent diagnostic algorithms are constrained by the limited sample size of the data and are prone to overfitting, resulting in limited applications. Conventional data augmentation is commonly adopted to expand the amount of training data, avoid overfitting, and improve the performance of deep models. However, in practical applications, random data augmentation, such as random image cropping or erasing, is difficult to realistically occur in specific tasks and may instead introduce tremendous background noises that modify actual distribution of data, thereby degrading model performance. In this paper, to assist in the early and accurate diagnosis of acute lymphoblastic leukemia, we present a ternary stream-driven weakly supervised data augmentation classification network (WT-DFN) to identify lymphoblasts in a fine-grained scale using microscopic images of peripheral blood smears. Concretely, for each training image, we first generate attention maps to represent the distinguishable part of the target by weakly supervised learning. Then, guided by these attention maps, we produce the other two streams via attention cropping and attention erasing to obtain the fine-grained distinctive features. The proposed WT-DFN improves the classification accuracy of the model from two aspects: (1) in the images can be seen details since cropping attention regions provide the accurate location of the object, which ensures our model looks at the object closer and discovers certain detailed features; (2) images can be seen more since erasing attention mechanism forces the model to extract more discriminative parts’ features. Validation suggests that the proposed method is capable of addressing the high intraclass variances located in lymphocyte classes, as well as the low interclass variances between lymphoblasts and other normal or reactive lymphocytes. The proposed method yields the best performance on the public dataset and the real clinical dataset among competitive methods.

## 1. Introduction

Acute lymphoblastic leukemia (ALL) is a neoplasm of precursor lymphoid cells and has high incidence worldwide, approximately 1.58 per 100 thousand individuals a year [1]. Its progress will be rapid and fatal if no clinical intervention is adopted. However, due to the development of treatment and supportive care in recent decades, the 5-year event-free survival rate has approached 90% in pediatric ALL but is still poor in adults [2,3]. According to the treatment guidelines released over the years, the precondition of further improving the survival rate is an accurate early diagnosis plus precise stratification of molecular biology and genetics regardless of what kind of innovative therapeutic approaches are taken [4].

ALL is characterized by the proliferation of lymphoblasts in bone marrow, peripheral blood, and vital organs in light of the French American British (FAB) classification standard [5,6,7]. Lymphoblasts are immature lymphocytes, and their overproduction controls bone marrow hematopoiesis function or causes hemorrhage, infection, and anemia. Manual morphological analysis of peripheral blood smears by microscopy is an important tool to assist in clinical screening for ALL, however, which is not cost-effective or convenient. This approach requires well-trained hematopathologists to invest adequate time and energy in screening blood samples and in identifying suspicious abnormal lymphoblasts. According to the International Council for Standardization in Hematology (ICSH) recommendations, at least 200 nucleated cells should be analyzed for each blood smear [6]. The manual inspection workload is tedious and repetitive. It is difficult to avoid mistakes caused by eyestrain and human negligence and to avert inconsistencies due to individual subjective decisions. The high requirements for the background and experience of hematopathologists make it difficult to achieve early and accurate diagnoses in hospitals located in remote regions. In addition, compared with other acute leukemias, the onset of ALL is more insidious and is easy to miss during manual microscope screening because it sometimes accompanies the normal blood cell count. Therefore, people are seeking an artifact intelligence-based method to solve the problem, relieve the working pressure on hematopathologists, and promote online consultation.

The entire workflow of blood smear inspection consists of smear preparation, cell detection, morphological analysis, and differential counting. Pathologists identify lymphoblasts based on a series of morphological characteristics, such as shape, nuclear chromatin, cytoplasm, and the presence of nucleoli or sometimes vacuoles. Recently, these steps have been partially solved through commercial hematology analyzers such as the SP1000i from Sysmex or CellaVision. Although there are already many automated blood cell analyzers in clinical laboratories that can alert or even classify blasts, it is still hard to further distinguish whether they originate from the myeloid or lymphoid system. In laboratories, there have been many attempts to develop automatic classification models to categorize cells based on their cell lineage. In a traditional machine learning-based workflow, after obtaining digital microscopic images of peripheral blood smears, there are several steps to generate the final classification, including preprocessing, segmentation, feature extraction, and prediction. Following this idea, research achievements have been proposed in recent decades. The most popular methodology concatenates color feature-based segmentation, texture feature extraction, and support vector machine-based classification into a workflow [7,8]. These methods utilize existing handcrafted features from computer vision, but it is difficult to comment on their performance since the features are designed for general images. In recent years, convolutional neural networks (CNNs) have become popular in solving image classification problems. Inspired by achievements in the natural image processing field, CNN-based methods are used to help diagnose ALL through peripheral blood or bone marrow samples [9,10,11]. Most of these methods utilize the existing classification network architecture through transfer learning to fit their blood sample images. From the medical point of view, it is important to understand the classification evidence, but unfortunately, the features generated by transfer learning are hard to interpret.

In general, the overall analysis of full blood cells needed in the early diagnosis of ALL is to classify lymphoblasts as well as normal leukocytes. To demonstrate this clearly, we list two datasets that consist of microscopic images of lymphoblasts and normal leukocytes from peripheral blood smears in Figure 1. C-NMC is the dataset released by ISBI in 2019 [12], and PD is the clinical dataset collected by our team (All input data are freely available from public sources). The precise classification of lymphocytes is nontrivial even for experienced hematopathologists. To investigate the reason, we drew the two-dimensional (to reduce memory consumption and runtime) distribution of data through the openTSNE toolbox [13] to observe data characteristics, as shown in Figure 2, where the green points and orange points refer to lymphoblasts and normal leukocytes, respectively. T-SNE [14] is a nonlinear dimensionality reduction algorithm that converts distances into conditional probabilities to express the similarity between points. It is very suitable for downscaling high-dimensional data to two- or three-dimensional planes for intuitive display to characterize whether the dataset is well separable (i.e., high interclass variances and low intraclass variances). Figure 1 and Figure 2 show that the cell classification is a typical fine-grained classification problem: (1) Low interclass variances: two populations of points with different colors mingle together, which indicates that lymphoblasts and normal white blood cells are highly similar apart from some minor differences. (2) High intraclass variances: points with the same color separate from each other, proving that there is a large variance in cells of the same type. Clinically, hematopathologists usually read the variance from typical discriminative parts of cells, such as coarse granularity in nuclei, chromatin tissue, and vacuoles, rather than from the entire structure. (3) Limited training data: although high-level CNN features possess good generalizability in representing images, the performance improvement is marginal under limited data. However, for hematopathologists, labeling each cell usually requires considerable time, which is an obstacle to enlarging the dataset.

The optimization of networks is based on batch of images, and all the images contribute equally during the optimization process. When the features of two samples from separated categories are approximately similar, or the features of two samples from same categories are extremely distinctive, the existing CNN cannot effectively separate them in the feature space, as shown in Figure 1 and Figure 2. Because of the these reasons, directly applying CNN models designed for the coarse-grained classification task, such as ResNeXt [15] and SENet [16], to leukocyte classification makes it difficult to obtain accurate classification results. In this study, we propose a weakly supervised ternary stream data augmentation fine-grained classification network to help diagnose ALL early from peripheral blood smears. Our main contributions are as follows:

We propose weakly supervised attention learning to generate attention maps (as shown in Figure 3) to extract the discriminative local features and locate the discriminative object’s parts from lymphocytes to help with early ALL fine-grained visual classification problem.

To improve the efficiency of data augmentation without modifying the actual distribution of data, we propose attention-guided data augmentation based on attention maps, including attention cropping and attention erasing. Attention cropping randomly crops and resizes one of the attention parts to present the details of distinguishing parts more effectively. Attention erasing randomly erases one of the attention regions out of the image to encourage the model to extract more features from other neglected discriminating regions.

Extensive experiments are conducted based on a public dataset to prove the advancement of the proposed framework compared with state-of-the-art methods and on a realistic clinical dataset to prove its promise in clinical practice.

## 2. Materials and Methods

In this section, we describe the proposed WT-DFN in detail, including weakly supervised attention learning and attention-guided data augmentation. The flowchart of the entire framework is illustrated in Figure 3. Our approach can direct the model to learn more discriminative semantic features (“see details”) while guiding it to discover some secondary features that are of complementary value for image recognition (“see more”), which can ultimately produce fine-grained discriminant features.

### 2.1. Embedding Weakly Supervised Attention Learning

To guide coarse-to-fine attention learning, we adopt weakly supervised learning to predict objects’ location distribution only by their category annotations. We extract the feature of image I∈ℝH×W (H is the height of the image, W is the width of the image) by the last layer of CNN backbone (before fully connected layer) and denote X∈ℝC×h×w as feature maps, where h, w, and C are height, width, and channel dimension (i.e., number of filters). The distribution of objects’ parts is represented by attention map A(·), which is obtained from X by Equation (1).
(1)F=f(X)=∪k=1MFk
where f(·) is a convolution function (Conv). Fk represents one of the objects’ part or visual pattern, such as nuclear chromatin, cytoplasm, or vacuoles. M is the number of attention maps. After representing object parts by attention maps F(·), we utilize bilinear attention pooling (BAP) [17] to extract features from these parts. We elementwise multiply feature maps Fk by each attention map X to generate a part matrix.

For each training image, we randomly choose one of its attention maps F(·) to guide the data augmentation process and normalize it as a kth augmentation map Fk*∈ℝh×w.
(2)Fk*=Fk−min(Fk)max(Fk)−min(Fk)

$F_{k}^{*}$ is used to guide the data augmentation. Based on $F_{k}^{*}$, cropping and erasing are performed in sequential to guide efficient data augmentation.

### 2.2. Attention Cropping Stream

Attention cropping stream aims to focus on the region of interest to see details features from discriminative parts, which is defined by the normalized attention map as shown in Equation (3). Concretely, we first obtain the crop mask using $F_{k}^{*}$ by setting element $F_{k}^{*}\left(i,j\right)$, which is greater than threshold 0.5 to 1, and others to 0, as shown in Equation (3).
(3)$$AB_{k}\left(i,j\right)=\{\begin{array}{l} 1,\;\mathrm{if}\;F_{k}^{*}\left(i,j\right)\geq 0.5\\ 0,\;\mathrm{otherwise}. \end{array}$$

The smallest bounding rectangular box that covers AB is cropped and denoted as $AB^{*}$. Next, we crop the corresponding $AB^{*}$ region on image I and zoom in this part’s region using bilinear interpolation algorithm (same size as the input image I) to obtain $I_{crop}$ to extract more details local feature, which is beneficial for solving fine-grained visual classification problem with high intraclass variance. As illustrated in Figure 4, since the scale of region of interest (ROI) of the object’s part increases, our model can look at the object closer and extract more fine-grained features.

### 2.3. Attention Erasing Stream

Attention erasing stream focuses on the less discriminative area to encourage model to propose multiple secondary information regions. Specifically, we first obtain attention erase mask using $F_{k}^{*}$ by setting element $F_{k}^{*}\left(i,j\right)$, which is greater the threshold 0.5 to 0, and others to 1, as represented in Equation (4).
(4)$$AE_{k}\left(i,j\right)=\{\begin{array}{l} 0,\;\mathrm{if}\;F_{k}^{*}\left(i,j\right)\geq 0.5\\ 1,\;\mathrm{otherwise}. \end{array}$$

Next, the $k_{th}$ part region will be erased by masking image I with $AE_{k}$ to obtain $I_{erase}$. One example of the cropping–erasing strategy is shown in Figure 4. Since the saliency map in $k_{th}$ part region is eliminated from image I, the model will be encouraged to propose other secondary discriminative parts, which will promote the model to see more and facilitate the solution of fine-grained visual classification problem with low interclass variance.

With attention-guided data augmentation, we obtained ternary stream from an input image, namely, the raw image and the two wrappings; cropped region of interest (ROI); and erased remainder, which correspond to the blue, orange, and green subpaths listed in Figure 4. In the first stage of the training phase (Phase-I), the raw image first generates whole image predictions and corresponding attention maps via the network. During the process, only the blue stream is activated, while the rest remain silent. Then, the attention map is used to extract the cropping ROI and the erasing remainder to wake up the orange and green streams and start the second training stage (Phase-II). The two stages iterate until convergence.

### 2.4. Voting Strategy

Due to attention-guided data augmentation, the classification of one input image can be achieved through the raw image, the cropped ROI, and the erased remainder. There is a chance that the three versions will sometimes generate different predictions. This might not be an issue in the training phase since the bias can be adjusted in the next iteration. However, it will become confusing if the same thing happens in the test phase. To address the problem, we develop the voting strategy (VS) algorithm, which uses a voting scheme to obtain the final prediction, as follows (Equation (5)):(5)$$p_{I}=\Gamma \left(p_{I}^{raw},p_{I}^{crop},p_{I}^{erase}\right)$$
where function Γ(·) returns the plurality of the prediction label for $p_{I}^{raw}$, $p_{I}^{crop}$, $p_{I}^{erase}$. An illustration of the selection algorithm is shown in Figure 5.

We obtain the final classification result by executing the above steps in turn. The detailed process of the proposed WT-DFN is described in Algorithm 1 (The algorithm flow of WT-DFN).
**Algorithm 1.** Coarse-to-Fine Prediction.**Require:** Trained WT-DFN model W**for** i=1,2,…,b **do** (batch size =b)1: Obtain raw probability piraw:piraw=W(Iraw,i)and out attention maps F by Equation (1);2: Calculate normalize Fk* by Equation (2);3: Obtain crop image Icrop and erase imageIerase by Equations (3) and (4), respectively;4: Predict fine-grained probabilitypicrop:picrop=W(Icrop,i),pierase:pierase=W(Ierase,i);5: Calculate the final prediction pi by Equation (5);**end for****return** {pi,…,pb}


### 2.5. Dataset and Evaluation Metrics

**Ethics statement**: This retrospective patient study was approved by the Institutional Review Board (IRB) of Zhongshan Hospital, Fudan University. All methods in this study were conducted in accordance with the relevant guidelines and regulations.

**Dataset**: We compare our method with the state-of-the-art methods on two datasets: the C-NMC dataset and the PD dataset. For the sake of fairness, we validate our proposed method mainly using the C-NMC dataset, which contains cell images from 84 cancer and 70 normal subjects. This dataset is released in three phases: (1) Phase-I. The malignant and normal cell numbers were 7272 and 3389 for 47 cancer subjects and 26 normal subjects, respectively. (2) Phase-II. A total of 1219 malignant cells from 13 cancer subjects and 648 normal cells from 15 normal subjects were released. (3) Phase-III. A total of 2586 cell images without labels from 9 cancer subjects and 8 normal subjects were released for online validation. All cell images are preprocessed via a stain-normalization procedure and a cell automated segmentation algorithm [12,18,19]. Each image has a resolution of 450×450 pixels and contains only a single cell. The PD dataset consists of 1478 images of acute lymphoblasts and 855 images of normal white blood cells collected at Zhongshan Hospital, Fudan University. All cells were segmented from the microscopic images and labeled by experienced hematologists following the standard clinical protocol of Zhongshan Hospital. For our experiments, ∼4/5 of the images of each class were assigned to the training set, and the remaining ∼1/5 were assigned to the test set to evaluate the performance of different architectures.

**Evaluation metrics**: For performance metrics, we adopt the weighted F1-score (WF1S) and area under the receiver operation curve (AUC). We denote TP as the true positives, FN as the false negatives, TN as the true negatives, and FP as the false positives. We refer to the number of categories in a given dataset (C-NMC or PD) as c and the total number of samples belonging to category i (i∈c) as ${\mathrm{SM}}_{i}$. F1S is defined as the harmonic mean of precision and recall. WF1S is the average weighting of each F1S category, as follows (Equations (6)–(8)):(6)$$F1S=\frac{2\cdot \mathrm{TP}}{2\cdot \mathrm{TP}+\mathrm{FP}+\mathrm{FN}}$$
(7)$$\mathrm{WF}1S=\frac{\sum_{i\in c}{\mathrm{SM}}_{i}\cdot {F1S}_{i}}{\sum_{i\in c}{\mathrm{SM}}_{i}}$$

Accuracy (acc) represents the ratio between the correctly predicted instances and all instances in the dataset:(8)$$\mathrm{acc}=\frac{\mathrm{TP}+\mathrm{TN}}{\mathrm{TP}+\mathrm{TN}+\mathrm{FP}+\mathrm{FN}}$$

## 3. Results

### 3.1. Experimental Setting

**Implement details:** The implementation of the proposed framework is mainly based on the open-source Python library PyTorch [20]. We train our model on a desktop computing workstation running on an Intel Core i9-7900× CPU @ 3.3 GHz (10 CPUs), 64 GB of DDR4 RAM, and one GeForce GTX 1080Ti, programmed with Python 3.6. In image classification experiments, the default hyperparameters are as follows: the training epochs are 120 in the rest of the paper unless otherwise specified; the batch size is 7 by default; the Ranger optimizer [21] is adopted with weight decay 1e-4; the initial learning rate is set to 0.01, with polynomial decay of 0.95 after every 2 epochs; we use linear learning rate warm-up, and the warm-up epochs are 2; we fix the input resolution r=300 unless otherwise noted. We also use “early stopping” [22] to prevent overfitting. The loss function we use is the cross-entropy loss function. In the process of validation, five-fold cross-validation is adopted and report means and variances of the five-fold cross-validation. All our experiments use the same hyperparameters as the default setting and start training from scratch unless otherwise specified.

### 3.2. Comparisons with State-of-the-Art Data Augmentation Works

We compare our proposed method with state-of-art data augmentation methods. For the fair comparison, we choose the C-NMC dataset for validation and the popular ResNet50 [23] as the backbone network for classification in the same way as [24]. To expand the data sample in a realistic way without introducing significant background noise, all input images were preprocessed with standard random left–right flipping, up–down flipping, affine transformation, grayscale transformation, and color jitter prior to any augmentations. Random erase (RE) [25] puts emphasis on simulating object occlusion issues, which can randomly select the rectangle region in an image and fill in a complementary value of zero. MixUp (MU) [26] uses two images to multiply and superimpose with different coefficient ratios and adjusts the label with these superimposed ratios simultaneously. CutMix (CM) [27] covers the cropped image to rectangle region of other images and then adjusts the label according to the size of the mix area. AugMix (AM) [24] utilizes stochasticity and diverse augmentations and a formulation to mix multiple augmented images to reduce the distribution mismatch problems encountered during training and testing. Compared with AutoAugment [28], RandAugment (RA) [29] narrows down the search space, which reduces the training complexity and may substantially reduce the computational cost. CutOut (CO) [30] is a simple regularization technique of randomly masking out square regions of input during training, which exhibits a certain commonality with RE. GridMask (GM) [31] is based on information dropping policy for data augmentation, which deletes uniformly distributed areas and finally forms a grid shape.

In Table 1 we show the comparison results on C-NMC. As for our method, our proposed WT-DFN achieves a superior accuracy of 83.41% without any complicated modules or external data, which bring about improvements of 0.95 points over the high baseline. Notably, when RandAugment and AugMix data augmentation are introduced during the training process, a significant performance degradation occurs, which we speculate is due to a dramatic change in the original distribution of the data. Finally, we also observe a minor decrease in the standard deviation of the model classification predictions when WT-DFN is introduced. Compared with the other data augmentation methods, WT-DFN outperforms the CutOut at least 0.42 points and achieves the state-of-art performances on C-NMC, which further demonstrates that WT-DFN is more suitable for fine-grained lymphocyte image categorization.

### 3.3. WT-DFN Performance on ResNets

To verify the consistency of WT-DFN’s performance improvement on CNN models in few-sample ALL image recognition, we chose widely available ResNets [23] networks of various depths to evaluate their performance on classification tasks in order to make the model fully and efficiently learn the images obtained through WT-DFN. During the train phase, we constructed a ternary stream network structure consisting of Raw stream, Crop stream, and Erase stream, as shown in Figure 3. The ternary stream classification model shared a CNN for feature extraction.

**ResNets.** We further verified the effectiveness of our approach on one popular residual network, ResNets [23]. Figure 6 presents the main results of our experiments. We make the following three observations: First, WT-DFN consistently brings significant performance improvements across different depths. In particular, WT-DFN achieves a ∼1.15% gain over ResNet-18 without introducing a substantial number of parameters. Second, as the depth of the network increases, the effectiveness of WT-DFN on the model performance gain gradually decreases. Last but not the least, ResNet-101 performs worse than ResNet-50 at the cost of a significant number of parameters, which is probably because the deeper neural network causes an overfitting problem and makes the gradients hard to propagate.

### 3.4. Comparison of the Proposed Classification Model with Other Methods

We compare the proposed method with other popular CNN-based methods to validate its effectiveness. The models we compare are fine-tuned on the C-NMC from their initial parameters trained on ImageNet to ensure their effectiveness, as shown in Table 2. There is one column listed in the table: final WF1S. The final WF1S is obtained by using the data from Phase-I and Phase-II as a training set and the data from Phase- III as a test set. Thanks to the fact that WT-DFN improves the automatic feature learning ability of the model directly from the representational capability when training from scratch, we outperform the remaining methods in the final WF1S. For the purpose of fair comparison, it should be noted that we chose the most extensively used model at the ISBI-2019 challenge, namely the SE-ResNeXt [16] network, as our backbone network. Actually, in the same experiment configuration as mentioned above, if we employ the initial parameters of SE-ResNeXt trained on ImageNet and fine-tune them on C-NMC, the best WF1S can reach 92.34% (https://competitions.codalab.org/competitions/20395#results (accessed on 17 September 2021)), which is obtained from the official evaluation server. Our method achieves significant improvement, demonstrating that the features produced by CNN can generate more discriminative representations through our ternary stream strategy.

**Training skills:** To reduce the significant distribution shift between the training and testing regimes, we apply FixRes [32] to the SE-ResNeXt [16] architecture. It is a fine-tuning strategy that retrains the classifier or a few top layers at the target resolution during only a few epochs.

**Table 2 diagnostics-12-00016-t002:** Comparison with state-of-the-art methods on the C-NMC final testing dataset.

Methods	Final WF1S(%) *Trained on Phase-I and II* *Tested on Phase-III*
MobileNetV2 ^+^ [33]	89.47
SE-ResNeXt ^+^ [34]	88.91
SENet ^+^ [35]	87.9
InceptionV2 ^+^ [36]	87.6
Inception ResNetV2 ^+^ [37]	84.8
NCA ^+^ [38]	91.04
DeepMEN ^+^ [39]	88.56
LSTM-DENSE ^+^ [40]	86.6
Stacking ^+^ [41]	85.52
WT-DFN	**92.30**

^+^ indicates methods with the initial weights learned on ImageNet in the relevant paper.

### 3.5. Convergence Analysis

We compared the training curves of our method with the baseline (SE-ResNeXt-50), as shown in Figure 7. SE-ResNeXt-50 has the same architecture as that proposed in the original reference and trained in single-stream mode, which means that the features extracted from the raw image are directly used to generate classification [16]. Based on the training curve shown in Figure 7, the proposed model achieves a lower training error after approximately 50 epochs. A similar result can be observed from the testing curve in Figure 7. The lowest testing errors of the baseline model and the proposed model are 13.96% and 12.61%, respectively. Compared to its plain counterpart, our method reduces the top-1 testing error by 1.35%, resulting from the successfully reduced training error. This indicates that the proposed ternary stream model has better generalization ability. We also note that the baseline converges faster than our methods because it is trained without the multistream strategy but attains a higher error rate (13.96%) on the test set. We speculate that the current Ranger solver cannot find good solutions to the plain net. In this case, under the complementary effects of the three streams, our method can continue to optimize the model.

### 3.6. Experimentation on the Clinical Dataset

To validate the effectiveness of our approach in practical clinical applications, we conducted experiments on a clinical dataset (PD). All settings followed those used in C-NMC, and five-fold cross-validation was employed. We also evaluated state-of-the-art classification models, including ResNet [23], SE-ResNeXt [16], MobileNetV2 [42], and Inception [43]. We compared our method with state-of-the-art methods on PD classification dataset. The comparison result on PD dataset is shown in Table 3. It can be seen that all models with WT-DFN achieved the state-of-art WF1S and accuracy. We significantly improved the WF1S and accuracy compared with light-weight architectures (Resnet18, MobileNetV2, and InceptionV1). Our proposed WT-DFN achieved WF1S of 80.50%, 85.52%, and 91.50% on PD dataset without external data, which bring about improvements of 22.53, 31.8, and 12.51 points over the baseline models. The performance meets the demand of clinical practice, and the small size of the model leads to potential embedding into medical equipment.

### 3.7. Ablation Studies

We studied the influence of the number of attention maps (A(·)) on the C-NMC dataset, denoted as M. Notably, we employed our proposed algorithm without FixRes in the experiment. When M reaches 32, the performance gradually reaches its peak, and the final WF1S score reaches approximately 87%, as shown in Table 4. However, as the parameter M increases, it requires more time to train the model, from 17.78 m to 18.40 m per epoch (increasing 3.48%). We consider the trade-off between training time and accuracy (normalized as if it is run on 1 GPU) and set M to 32 to achieve the best results.

## 4. Discussion

The CNN proposed in this study shows excellent performance in identifying key pathological cells when both malignant leukemia cells and normal cells are present in peripheral blood samples. We present an effective method for accurate, automatic recognition of ALL cells from blood smear microscopic images and perform extensive experiments on the C-NMC dataset to investigate its effectiveness. The C-NMC dataset attains WF1S values of approximately 92.30%, allowing these cells to be identified with very high accuracy, which is superior to other sophisticated classifiers in the literature [32,39,40].

We also compiled a dataset of approximately 2333 practical clinical single-cell images of morphologies relevant in the diagnosis of ALL from the peripheral blood smears of multiple individuals. After annotation by hematopathologists, we used this dataset to train and evaluate state-of-the-art CNN-based classification models. The network shows promising performance in distinguishing the morphological cell types and achieved outstanding accuracy (approximately 91.90%) in this dataset. Considering this superior performance and the fact that our method is scalable and amenable to end-to-end training, the proposed algorithm can be utilized to rapidly assess thousands of cells on a blood smear scan, which helps pathologists locate suspicious cells more effortlessly, especially when the number of malignant cells is small (e.g., in the early stages of ALL or the initiation of relapse), which is precisely the situation where manual microscopy is the easiest to miss and time consuming. Without adjusting the entire workflow (smear preparation, cell detection, morphological analysis, and differential counting) of the existing clinical protocol, the WT-DFN model could be easily integrated as a step of existing computer-aided diagnostics (CAD) to assist medical personnel in cytomorphological analysis and facilitate early detection of leukemia. It contributes to providing real-time analysis and better decision-making tools.

Although the proposed method achieves quite promising results, there are still some limitations: (1) Compared with other existing CNN methods, the proposed framework requires more training time in the training phase (about 3×). However, our method can be trained end-to-end without other manual annotations except for category labels and needs only one network with one propagation during testing. (2) VS uses the combination of coarse-to-fine probabilities to predict the final labels instead of directly using the probabilities calculated by the one-way CNN forwarding process [44]. Our voting strategy works only during the test phase, which slightly increases the computational burden. (3) We anticipate that further expansion of the dataset will improve the network classification performance. It will be interesting to observe how our presented method performs on other data, which is also another future work in our group. Finally, the experts who provide the ground-truth labels for single-cell images can perform the screening on the entire smear images and therefore compare the cellular patterns present from a global point of view. Instead, our method is forced to make a classification decision based on a single-cell image alone, without the ability to compare to other cells from the same patient, which may be solved by embedding our method into an object detection framework in the future.

## 5. Conclusions

In this paper, we proposed a ternary stream fine-grained classification model to distinguish lymphoblasts from normal white blood cells and reactive lymphocytes based on microscopic images of peripheral blood smears. It aims to help the early diagnosis of acute lymphoblastic leukemia and possesses the potential to serve as a rapid prescreening and quantitative information decision-making tool for cytologists. The auxiliary diagnostic capability of the method might be further enhanced when integrated with other intrinsic quantitative methods developed for diagnosing hematological malignancies, such as flow cytometry or molecular genetics. Future research includes evaluating our framework on more public datasets and clinical datasets and promoting its application in clinical practice. The code will be released after the article is accepted (https://github.com/YunDuanFei/WT-DFN accessed on 17 September 2021).

## Figures and Tables

**Figure 1 diagnostics-12-00016-f001:**
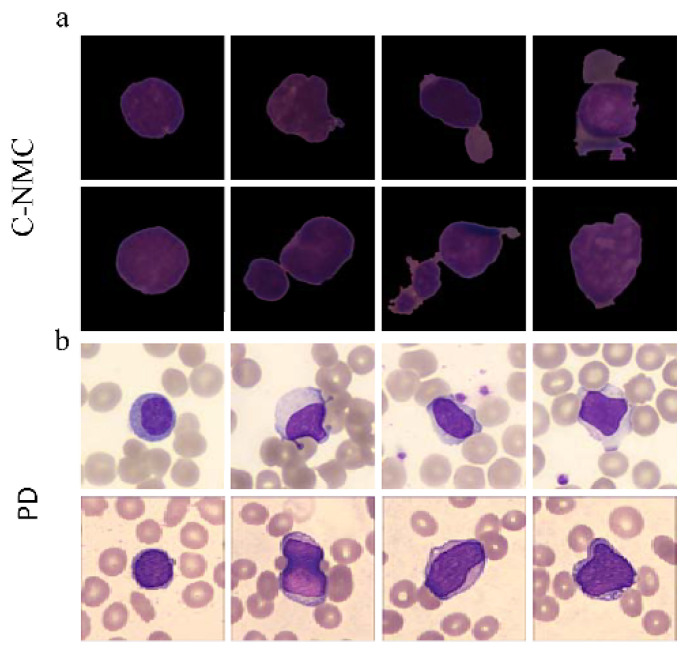
Cell image samples from the dataset. (**a**) C-NMC dataset: row #1 shows lymphoblasts, and row #2 shows normal leukocytes; (**b**) PD dataset: row #1 shows lymphoblasts, and row #2 shows normal leukocytes. The classification of lymphocytes is nontrivial even for experienced hematopathologists.

**Figure 2 diagnostics-12-00016-f002:**
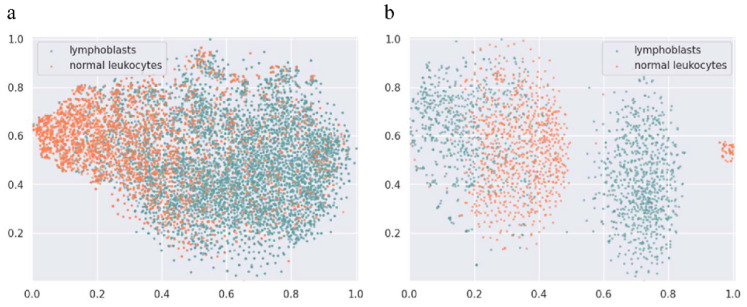
OpenTSNE visualizations, where the green and orange points represent lymphoblasts and normal leukocytes, respectively. (**a**) C-NMC dataset (sample size=10661); (**b**) PD dataset (sample size=2333). Cluster labels are not shown for visual clarity. Figure best viewed in color.

**Figure 3 diagnostics-12-00016-f003:**
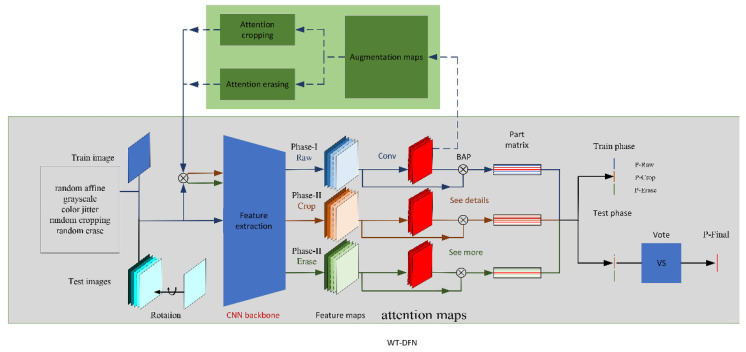
Flowchart of the proposed ternary stream fine-grained classification model. BAP denotes bilinear attention pooling.

**Figure 4 diagnostics-12-00016-f004:**
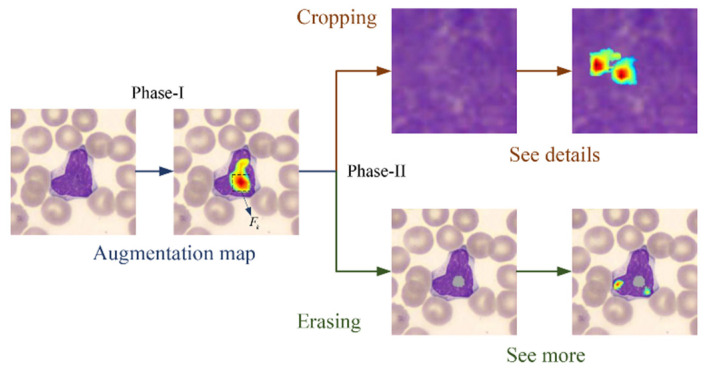
Attention maps represent discriminative parts of the object. The regions with high-attention coefficients are cropped and up-sampled to extract more detailed part features for “see details”. The remaining regions after erasing the regions with high attention are used to generate more discriminative object parts for “see more”.

**Figure 5 diagnostics-12-00016-f005:**
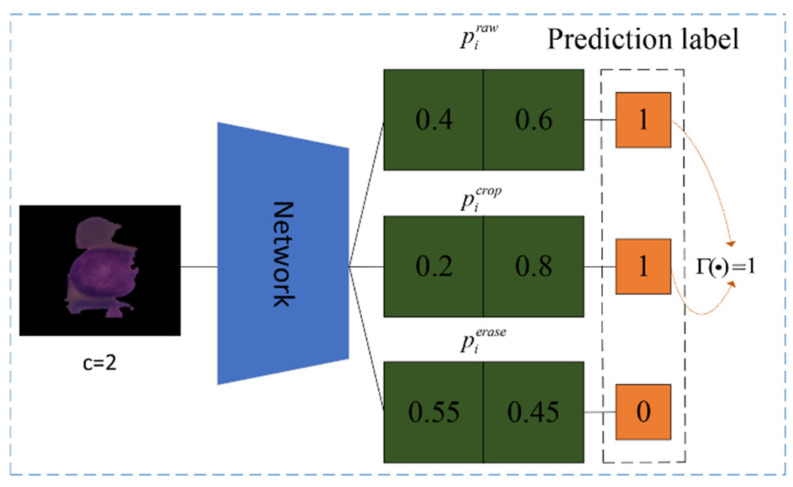
The example of the voting algorithm. The number of categories is c=2.

**Figure 6 diagnostics-12-00016-f006:**
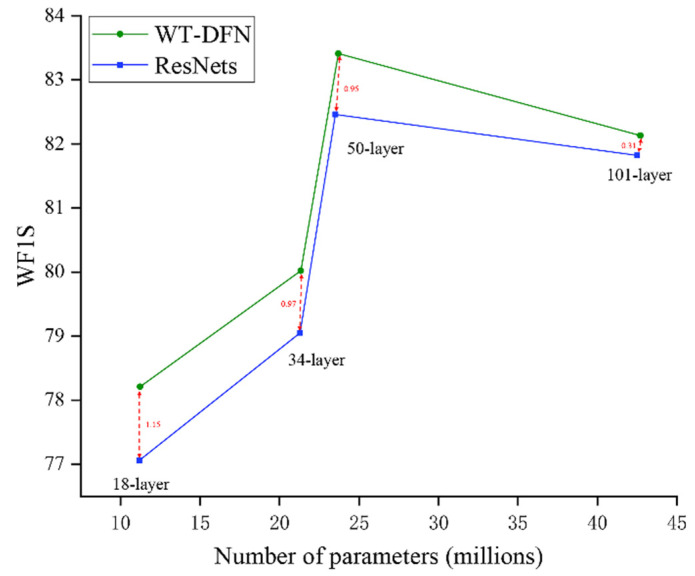
WF1S comparison on C-NMC. With similar number of parameters, our method consistently outperforms the baseline ResNet by a large margin.

**Figure 7 diagnostics-12-00016-f007:**
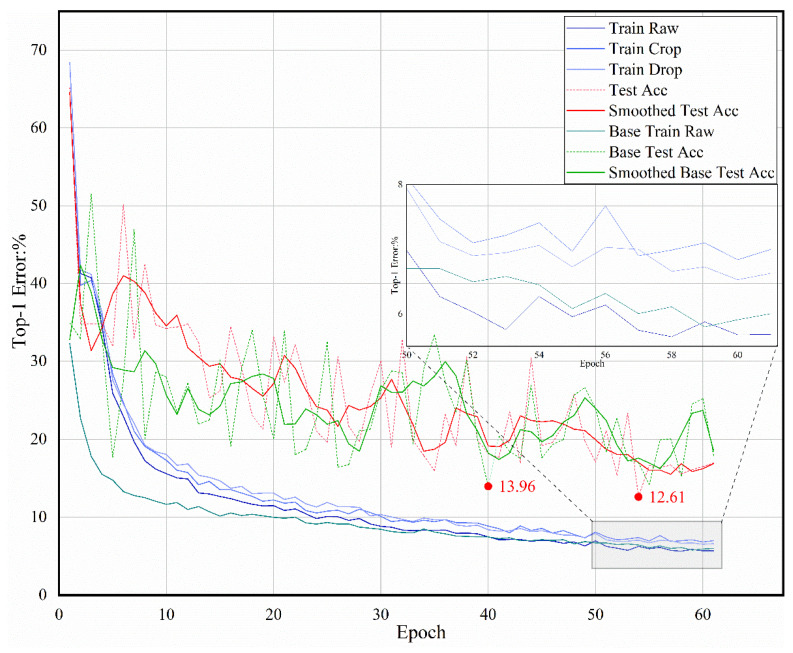
Training curves of our method and the baseline on C-NMC. Dashed lines denote the testing error. Our method performs better than the baseline.

**Table 1 diagnostics-12-00016-t001:** Comparisons with state-of-the-art augmentation methods on C-NMC.

Method	WF1S
	82.46±0.95%
**RE**	82.35±1.61%
**RA**	61.09±1.54%
**MU**	80.02±1.01%
**CM**	79.07±2.24%
**AM**	70.57±0.95%
**CO**	82.99±1.03%
**GM**	80.41±0.83%
**WT-DFN**	83.41±0.57%

**Table 3 diagnostics-12-00016-t003:** Comparison with state-of-the-art methods on PD.

Backbone	Without WT-DFN					With WT-DFN				
	Parameters	FLOPS	Top-1 acc	AUC	WF1S	Parameters	FLOPS	Top-1 acc	AUC	WF1S
ResNet18	11.18M	3.61G	59.38	51.78	57.97	11.23M	3.62G	**78.88**	**81.01**	**80.50**
ResNet34	21.29M	7.32G	62.17	62.08	62.53	21.34M	7.33G	**78.80**	**79.65**	**79.32**
SE-ResNeXt26	14.75M	4.95G	78.02	82.73	78.64	14.94M	4.96G	**85.33**	**87.00**	**85.65**
SE-ResNeXt50	25.51M	8.51G	77.01	82.75	77.56	25.71M	8.52G	**88.44**	**86.02**	**88.36**
MobileNetV2	2.23M	0.63G	66.96	50.00	53.72	2.35M	0.64G	**85.97**	**82.61**	**85.52**
InceptionV1	5.98M	3.16G	78.57	79.04	78.99	7.10M	3.16G	**91.43**	**91.30**	**91.50**
InceptionV2	13.47M	3.79G	67.41	72.76	67.10	13.57M	3.79G	**88.76**	**85.60**	**88.44**
InceptionV3	21.79M	5.91G	78.57	81.77	79.14	21.99M	5.91G	**79.87**	79.87	**80.32**
InceptionV4	41.15M	12.61G	90.49	90.52	90.72	41.29M	12.61G	**91.97**	90.05	**91.90**
Inception ResNetV2	54.31M	13.49G	82.26	78.53	81.14	54.46M	13.49G	**88.12**	**88.66**	**88.36**

**Table 4 diagnostics-12-00016-t004:** The effect of the number of attention maps evaluated on the C-NMC dataset.

M	Preliminary WF1S(%)	Total Time ^∆^ (m)
4	83.79	17.78
16	86.24	17.80
32	87.12	17.89
64	85.98	18.40

^∆^ Total time = Forward time + Backward time.

## Data Availability

The public dataset is openly available under reference [12]. The other part of the data presented in this study is available on request from the corresponding author. The data are not publicly available due to the ethical restrictions and privacy.

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
