# Peer review of "Weakly Supervised Ternary Stream Data Augmentation Fine-Grained Classification Network for Identifying Acute Lymphoblastic Leukemia"

_diagnostics, 2021, doi:10.3390/diagnostics12010016_

Round 1

Reviewer 1 Report

The paper introduces an original idea, well-planned investigation, and properly analyzed research results. 

Table 1 is presenting results in an unusual way. In my opinion, a single line or column with results could be a better option for presenting results. Also, numerical values in the text seem to be using smaller font sizes. It looks a bit strange. An additional figure in Figure 7 is non-readable.

The accuracy of the investigated model varies from epoch to epoch. The presented results, e.g. WF1S score is given in a very precise manner, showing a scale of 2 in the values. I wonder if we repeat once against the training of the model, after the same number of epochs, will we receive the same or very similar values. It would be good to discuss a little bit this point. Maybe the values, presented in the tables show the very best achieved results, o just the WF1S after the last epoch of training. If so, then there is a question for discussion: maybe it is worth adding one additional training epoch, or 10, or maybe 20. Otherwise, maybe it is not worth to use the scale of 2 in presented values.

Reviewer 2 Report

This is a manuscript reporting on ‘’Weakly Supervised Ternary Stream Data Augmentation Fine-Grained Classification Network for Identifying Acute Lymphoblastic Leukemia’’. In this paper authors propose a ternary stream fine-grained classification model to distinguish lymphoblasts from normal white blood cells and reactive lymphocytes based on microscopic images of peripheral blood smears and propose it as a complementary diagnostic method to well-established flow cytometry and molecular diagnostics They developed it by first generating attention maps to represent the distinguishable part of the target by weakly supervised learning, and then getting the rest two streams via attention cropping and attention erasing.

Broad comments: The manuscript is well written and reports on interesting matter, but there are several concerns that need to be addressed to improve the value of the manuscript.

Specific comments:

  1. The authors conducted their analysis on peripheral blood smears. It would be of pivotal interest to test their model on bone marrow smears since it is a gold standard for diagnosis of acute lymphoblastic leukemia. The authors state ‘’ The current clinical routine of the first step in the diagnosis of ALL is manual morphological analysis of peripheral blood smears through a microscope…’’ which is not true if you reflect on the guidelines by Brown PA et al. Acute Lymphoblastic Leukemia, Version 2.2021, NCCN Clinical Practice Guidelines in Oncology. J Natl Compr Canc Netw. 2021;19:1079-1109. that state ‘’ If there are sufficient numbers of circulating lymphoblasts (at least 1,000 per microliter as a general guideline) and clinical situation precludes bone marrow aspirate and biopsy, then peripheral blood can be substituted for bone marrow.’’ – so peripheral blood smears are not the first to evaluate if you have a suspicion of ALL in a patient.
  2. The authors mention FAB classification of ALL, which is an old one, please correct it and cite relevant literature on the new classification: Arber DA et al. The 2016 revision to the World Health Organization classification of myeloid neoplasms and acute leukemia. Blood. 2016;127:2391-405.
  3. Please cite an article by. Terwilliger, T., Abdul-Hay, M. Acute lymphoblastic leukemia: a comprehensive review and 2017 update. Blood Cancer J. 7, e577 (2017).
  4. There are generic sentences in the first paragraph of discussion, please delete.
  5. Please enlarge figures so they are easier to read, eg. I can hardly see anything on Figure 3.

Round 2

Reviewer 2 Report

I have carefully read the review version of the manuscript. The authors addressed issues 2-5, but issue 1 has not been yet addressed. I still think that it is important to show the function of the model on the sample of bone marrow aspirates, and not only peripheral blood. 

Round 3

Reviewer 2 Report

The authors have now managed to clarify the rationale for the investigation of peripheral blood smears. No further revisions are needed.